# Genome-wide insights into adaptive hybridisation across the *Schistosoma haematobium* group in West and Central Africa

**Toby Landeryou**[1,2,3]*, **Muriel Rabone**[2,3], **Fiona Allan**[2,3], **Rosie Maddren**[1,3], **David Rollinson**[2,3], **Bonnie L. Webster**[2,3], **Louis-Albert Tchuem-Tchuenté**[4], **Roy M. Anderson**[1,3], **Aidan M. Emery**[2,3]

**1** Department of Infectious Disease Epidemiology, School of Public Health, Faculty of Medicine, Imperial College London, London, United Kingdom, **2** The Natural History Museum, Department of Life Sciences, London, United Kingdom, **3** London Centre for Neglected Tropical Disease Research (LCNTDR), Department of Infectious Disease Epidemiology, School of Public Health, Faculty of Medicine, Imperial College London, London, United Kingdom, **4** Centre for Schistosomiasis and Parasitology, University of Yaoundé I, Yaoundé, Cameroon

* t.landeryou@imperial.ac.uk

**Data Availability Statement:** SRA data accession numbers for all the data used is the following: ☐ SRR17445877 ☐ SRR17445876 ☐

## Abstract

Schistosomiasis remains a public health concern across sub-Saharan Africa; current control programmes rely on accurate mapping and high mass drug administration (MDA) coverage to attempt disease elimination. Inter-species hybridisation can occur between certain species, changing epidemiological dynamics within endemic regions, which has the potential to confound control interventions. The impact of hybridisation on disease dynamics is well illustrated in areas of Cameroon where urogenital schistosomiasis, primarily due to *Schistosoma haematobium* and hybrid infections, now predominate over intestinal schistosomiasis caused by *Schistosoma guineensis*. Genetic markers have shown the ability to identify hybrids, however the underlying genomic architecture of divergence and introgression between these species has yet to be established. In this study, restriction site associated DNA sequencing (RADseq) was used on archived adult worms initially identified as; *Schistosoma bovis* (*n* = 4), *S. haematobium* (*n* = 9), *S. guineensis* (*n* = 3) and *S. guineensis x S. haematobium* hybrids (*n* = 4) from Mali, Senegal, Niger, São Tomé and Cameroon. Genome-wide evidence supports the existence of *S. guineensis* and *S. haematobium* hybrid populations across Cameroon. The hybridisation of *S. guineensis x S. haematobium* has not been demonstrated on the island of São Tomé, where all samples showed no introgression with *S. haematobium*. Additionally, all *S. haematobium* isolates from Nigeria, Mali and Cameroon indicated signatures of genomic introgression from *S. bovis*. Adaptive loci across the *S. haematobium* group showed that voltage-gated calcium ion channels (Ca$_v$) could play a key role in the ability to increase the survivability of species, particularly in host systems. Where admixture has occurred between *S. guineensis* and *S. haematobium*, the excess introgressive influx of tegumental (outer helminth body) and antigenic genes from *S.*

SRR17445875 ☐ SRR17325806 ☐
SRR17325805 ☐ SRR17325804 ☐
SRR17325803 ☐ SRR17325943 ☐
SRR17325942 ☐ SRR17325941 ☐
SRR17325940 ☐ SRR17325939 ☐
SRR17325938 ☐ SRR17325937 ☐
SRR17325936 ☐ SRR17325935 ☐
SRR17325934 ☐ SRR17325933.

**Funding:** The study was supported financially by the Wellcome Trust under grant number 104958/Z/14/Z to AME and DR. Salary was received from the Wellcome Trust (www.wellcome.org) grant number 104958/Z/14/Z for authors FA, TL and MR. The funders had no role in study design, data collection and analysis, decision to publish, or preparation of the manuscript.

**Competing interests:** The authors have declared that no competing interests exist.

*haematobium* has increased the adaptive response in hybrids, leading to increased hybrid population fitness and viability.

## Author summary

Current global anthropogenic changes have contributed to the increased transmission of infectious diseases. One of the direct consequences of these changes is the geographical distribution of schistosomiasis, resulting in the natural formation of hybrid populations. Hybridisation between *S. guineensis* and *S. haematobium* in particular has led to epidemiological variability where they exist sympatrically, changing the region endemicity from predominantly intestinal to urogenital schistosomiasis. This has occurred in part due to the replacement of *S. guineensis* through the invasive presence of *S. haematobium*. The underlying genomics behind hybrid viability and adaptation has yet to be explored within the species group. This study addresses this need through the analysis of genome-wide datasets, to highlight adaptive loci that could be beneficial to hybrid viability and increased fitness of *Schistosoma* hybrids.

## Introduction

Urogenital schistosomiasis is a waterborne parasitic disease transmitted through certain species of freshwater snail inhabiting water bodies throughout much of sub-Saharan Africa [1]. Infection results from contact with fresh water contaminated by the free-living cercariae stage of the *Schistosoma* life cycle. Across sub-Saharan Africa, 240 million people are affected by schistosomiasis, with *Schistosoma haematobium* causing urogenital schistosomiasis [2]. The *S. haematobium* species group comprises eight other sister species including; *S. intercalatum* and *S. guineensis* causing human intestinal schistosomiasis, and *S. bovis*, *S. mattheei*, *S. margrebowei*, *S. leiperi*, *S. curassoni* and *S. kisumuensis* infecting various wildlife, ruminants and livestock.

Due to the close phylogenetic relationships across the *S. haematobium* group, combined with overlapping host species ranges, inter-species hybridisation within the is not uncommon. Studies by Rollinson et al. [3] report laboratory crosses, whilst Webster et al. [4] detected field-derived hybrids from endemic settings [5,6]. The presence of hybrids is of particular importance when considering disease control, as current intervention policy indicates a shift from morbidity control to transmission interruption [7]. To make community elimination possible, it is important for MDA to achieve high levels of coverage [8], ensuring no reservoirs of infection remain that would continue to re-infect the treated community [9]. Hybridisation between members of the *S. haematobium* group introduces significant barriers to community elimination through solely human-based MDA interventions. The discovery of hybridisation between human and animal schistosomes: *S. haematobium x S. bovis* [10], *S. haematobium x S. mattheei* [11] and *S. haematobium x S. curassoni* [12] will increase the role that wild mammals and livestock have as animal reservoirs driving transmission, even when human prevalence reaches levels suitable for transmission interruption. Efficacy of praziquantel on hybrid and mixed infections within humans remains strong [13,14], hybridisation events would directly impact current monitoring, evaluation and mapping of the disease within endemic areas [15–17]. Laboratory experiments have demonstrated that $F_1$ and $F_2$ hybrids between *S. guineensis* and *S. haematobium* exhibited a greater ability to infect snail intermediate hosts along with

increased growth rate, reproductive potential and longevity [13–15]. The increased infection potential of hybrids across multiple species of freshwater snail host will increase hybrid populations ability to establish diseases within environments, increasing infection range [18,19].

Introgression refers to the exchange of genes between divergent lineages that result from backcrossing of hybrids with one or both parents. Also described as hybridization, this theoretically should result in a relatively even mixture of both gene and allele frequencies deriving from both parental species in early hybridisation events [20]. The genomes of hybrids are characterized by large amounts of genetic variation compared to single species parental counterparts. However hybrid genomes regularly contain novel genetic combinations that may alter functionality of certain genes [21]. Patterns of introgressive hybridisation in *Schistosoma* has typically been documented using a limited set of genetic markers [4,5,17,22]. As yet there is limited understanding of the dynamics of locus-specific introgressive hybridisation within the genus. This limitation has hampered the detailed understanding of the genomic dynamics of introgression, and the role it plays in hybrid populations of schistosoma parasites. Hybrid zones are regions where the genetic architecture of local adaptation or lack of reproductive isolation in schistosomes can be examined. These regions frequently juxtapose the underlying ecological gradients that govern the genetic make-up of a population, allowing for genome-wide quantification of selection [20,23]. Selection may prevent introgression at certain loci, underpinning the maintenance of speciation barriers conveying certain adaptive traits [24]. Alternatively, genome-wide introgression could impart functional genomic adaptation, where beneficial loci are inherited within hybrids increasing overall population fitness. Introgression has been implicated in the proliferation of an invadolysin gene allele (Smp_127030) in populations of *S. bovis* and *S. haematobium* hybrids, which could increase the infective success of hybrids [25, 26]. Examples like this highlight the importance of genome-wide data sets when investigating hybrid zones, allowing species boundaries and loci affecting them to be defined according to potential phenotypic adaptation. Species boundaries in the *S. haematobium* group have seen a significant erosion due in large part to climatic [27] and anthropogenic [3,10,28] factors, affecting the movement of intermediate host snail species via watershed changes and translocation and the proximity that definitive host interact.

Nowhere have the epidemiological ramifications of hybridisation in schistosomiasis been more evident than in Cameroon. Early evidence for hybridisation between *S. haematobium* and *S. guineensis* had been reported across several areas in Cameroon using predominantly the morphology of eggs found within infected individuals urine [13, 25–28]. Molecular analysis confirmed the presence of hybrids using *cox1* and nrDNA ITS2 + IGS markers [27,29,30]. Infection surveys revealed that within a period of around 25 years the population of hybrids declined over time, suggesting that *S. haematobium* had completely replaced the incumbent *S. guineensis* species making urogenital schistosomiasis the dominant infection [29,31,32]. Sanger sequencing of the targeted mitochondrial and nuclear markers produced sufficient molecular data to detect the presence and monitor hybrids in the sympatric regions in southwest Cameroon, but the region has not been studied for some time. Using more modern methods, which allow genome-wide sequencing would allow the analysis of underlying genomic characteristics of introgressive hybridisation, which could highlight potential adaptive traits that have made such a drastic change possible.

In this study, we address this need by examining historical field derived isolates collected from infected individuals and snails in Cameroon, Nigeria, Senegal, Mali and the islands of São Tomé. This dataset includes previously identified single species isolates of *S. bovis* from Senegal, *S. haematobium* from Cameroon, Nigeria and Mali, *S. guineensis* from São Tomé alongside admixed isolates of *S. haematobium x S. guineensis* from Cameroon. The aim of this study was to identify genome-wide patterns of selection within the *S. haematobium* group

across the region. Two questions were considered: 1) is there evidence of outlier loci providing differentiation between species in the region? And 2) what role does introgression play in the dominance of *S. haematobium* over *S. guineensis* in sympatric zones? To address these questions, we analysed restriction site associated DNA sequencing (RADseq) data to identify tens of thousands of genomic variations. The resulting dataset was used to characterise patterns of genomic ancestry across *Schistosoma* isolates of hybrid ancestry. This analysis was coupled with the use of genomic clines [33] to study patterns of introgression in hybrids to detect loci that deviate from the genome-wide expectation of introgression, highlighting regions and specific genes that would be beneficial to increased infection/immunomodulation of infected host.

## Methods

### Sampling

Samples used for sequencing were adult, male; schistosome worms due to DNA yield requirements for RADseq. Field collected schistosome isolates were transferred to the lab in infected freshwater intermediate snails, which were allowed to produce schistosome cercariae subsequently passaged through laboratory definitive rodent hosts to generate adult worms. Adult worms, stored long-term in liquid nitrogen (approx. -180˚C), were thawed and rehydrated for extraction and library preparation workflows. The long term passage of schistosoma worms within a laboratory setting can lead to increased selective advantage of specific genotypes. To mitigate this, samples used within this study were selected from worms derived from first passage; this ensures samples used are true representation of field-derived genotypes. Field isolates were acquired between 1983 and 2006 (S1 Table) from Nigeria, Cameroon, Senegal, Mali and the islands of São Tomé (Fig 1). The identification of hybrid worms were initially performed using egg morphology at the point of collection, and subsequent Sanger identification methods were performed using mito-nuclear combination sequencing to allocate isolates to specific species groups prior to downstream sequencing. This was particularly used to identify admixed isolates, as has been used in previous studies to confirm the presence of hybrid isolates across the *S. haematobium* species group.

### Genomic sequencing and bioinformatic processing

DNA was sequenced from a total of 18 worm pairs were separated, with DNA extracted from individual males [34,35,36]. DNA was extracted using QIAamp DNA Blood and Tissue extraction kit (Qiagen) following the manufacturer's instructions. DNA concentration of each extract was quantified using Qubit fluorometric quantification. Library preparation and RAD-seq preparation were performed by Floragenex sequencing facility as per *Sbfl* enzyme digestion guidelines [37].

PCR duplicates were removed from raw sequencing read data using clone_filter module [38] of Stacks v2.0. Reads were demultiplexed to individuals using the Stacks process_radtags module before trimmed paired end reads were assembled to create a "pseudo-reference" genome to map individual's datasets, using the dDocent RADseq variant-calling pipeline [39]. The *de novo* approach was favoured due to the addition of admixed individuals into the sample pool. *De novo* assembly generated 28,189 loci, with an average length of 196 bases. Individual read files were then mapped to the "pseudo-reference" genome with the BWA-mem [40] algorithm using an open gap penalty of three and a mismatch penalty of two. Two process mapped files SAMtools v1.3 [41] and BCFtools v1.3 [42,43] were used, with final SNP-calling performed using VCFtools [44,45], and a SNP calling quality score of 30. Reducing one sampled variant site from each locus reduced the effect of linkage disequilibrium between variant sites.

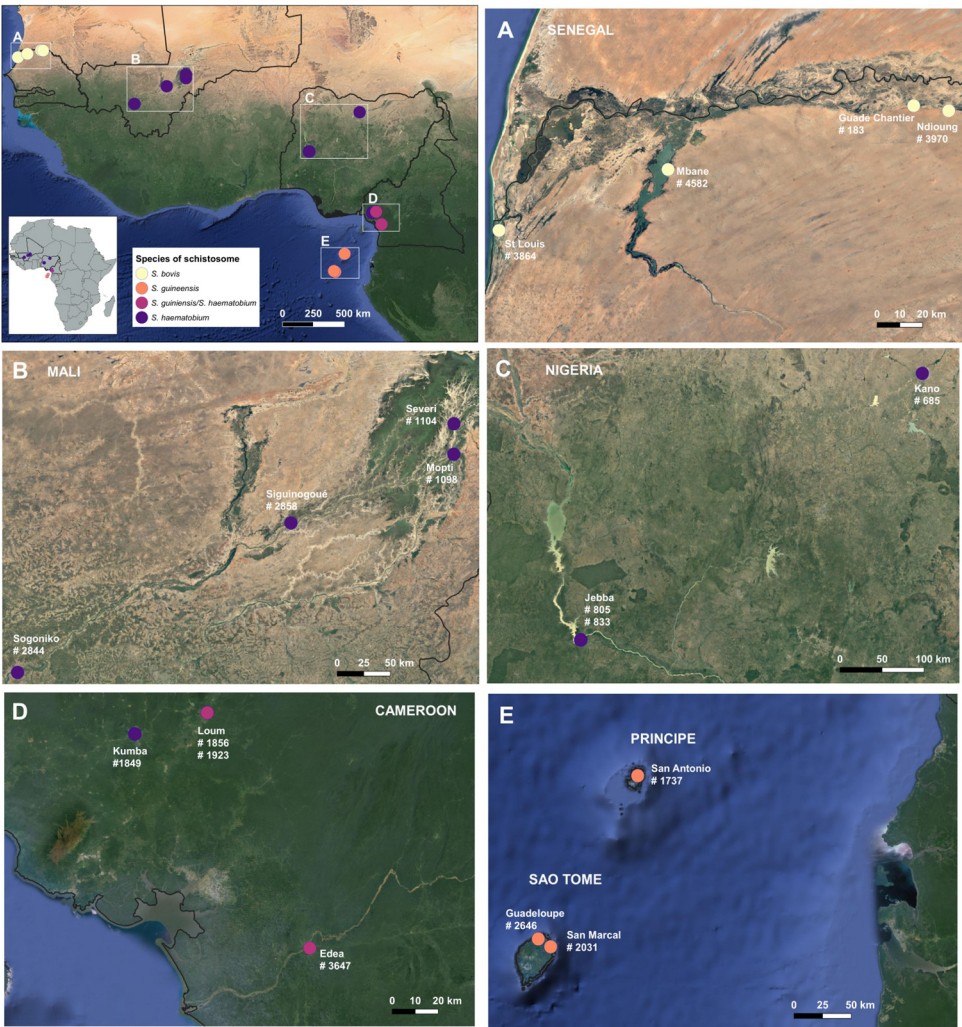

**Fig 1. Geographic locations and the initial species identification of the isolates analysed.** The map was created in Quantum GIS, version 3.14 (Pi), and the satellite layer sourced from 'openlayers' via the OpenLayers python plugin deriving from OpenStreetMaps Satellite data. Base-layer tileset were downloaded as follows: Mali Tile Bounds -8.30625, 12.45391, -3.97171, 14.69223; Senegal Tile Bounds -16.53588, 15.91597, -14.56542, 16.70207; Nigeria Tile Bounds 5.07816, 8.90230, 8.62072, 12.09504; Cameroon Tile Bounds 9.22728, 3.66772, 10.32132, 4.91165; Sao Tome and Principe Tile Bounds 6.07606, -0.47990, 9.78499, 1.98712; West African Tile Bounds -17.40185,-1.58269,14.02316,16.63042.

The calculation of population genomic statistics was performed according to methods described by Hohenlohe et al [46], using the Stacks tool, where the degree of polymorphism within populations with statistic nucleotide diversity $\pi$ and differentiation among populations with statistic fixation index $F_{st}$ was measured. The average $F_{st}$ across the whole-genome was performed using the kernel smoothing approach. Selecting loci with significantly elevated $F_{st}$ values between species and admixed populations identified SNP loci with elevated signatures of selection.

## Differentiation of *Schistosoma* species

To ensure accuracy of species identification, a phylogenetic analysis of unlinked SNP loci was created and converted in PHYLIP format using STACKS populations' workflow. Phylogenetic

inference was estimated from concatenated SNP data set using maximum likelihood (ML) in the programme RAxML v8.1.16 [47], using a GTR + G likelihood model of evolution. Likelihood calculations were corrected for ascertainment bias, which adjusts for invariable sites within the SNP dataset. Before analysis, the dataset was filtered using a custom script to exclude sites that displayed invariability ambiguity [48]. Support for the tree was assessed using 500 bootstrap replicates with out-groups established via concatenated RAD loci extracted from *S. mansoni* (GCA_000237925.4) and *S. rodhaini* (ERX092221) available SRA datasets.

A total of 10,000 loci were analyzed using STRUCTURE v. 2.3.4 [49]. For polymorphic loci that contained more than one SNP, only the first SNP was included to minimize the number of linked markers. Polymorphic loci that contained a SNP differentially fixed between species and outlier loci were excluded from further analysis, since these loci would be assumed to be under selection and did not represent neutral genetic differentiation between species isolates. Population differentiation was assessed using a model of admixture and correlated allele frequencies. A total of 500,000 Markov chain Monte Carlo (MCMC) repetitions were performed after discarding a burn-in of 100,000. We tested for up to five different genetic clusters ($K$ = 1–5) in four consecutive independent runs. To compare the likelihoods between different numbers of genetic clustering, $\Delta K$ was estimated [50]. Using the most likely number of clusters identified, admixture of each individual was assessed using the $Q$-value in STRUCTURE; this indicates the proportion of each genome that is assigned to each one of the pre-defined genetic clusters. Additional population structuring was investigated using a Principal Coordinate Analysis (PCoA) on the same set of polymorphic loci, using the program GENALEX v. 6.5 [51] and further plotted using Adegenet R software package [52]. Genetic variance within and between clusters was performed using the AMOVA $F_{st}$ implemented in GENALEX, using 1,000 permutations.

## Analysis of genomic introgression

We conducted Bayesian estimation of genomic clines using the program bgc [36] to identify loci that defied the natural expectation of admixture compared to the genomic diversity background signal. The samples were segregated between parental species of *S. haematobium*, *S. guineensis* and admixed hybrid isolates, as these were two parental species contributing to the admixed individuals. Bayesian genomic clines estimates the hybrid index ($h$) for each of the individuals that are within the admixed population and establishes the proportion of an individual's genome/specific loci from one parental population. Using two locus-specific genomic cline parameters; α, the genomic cline centre parameter and β, the genomic cline rate parameter, established hybrid indexes. Both values are used to establish the posterior probability of inheritance from one parental population at a specific locus within the admixed population. If both α and β are equal to 0, then neutral genomic background expectation is suggested, this is established with a dashed line across the cline plot. Loci that are expected to have excess ancestry from one of the parental populations will be identified based on each cline parameter and classified based on their locus-specific cline parameter relative to the genome-wide cline distribution [53]. To establish differential effects of excess introgressive potential within admixed isolates, loci were divided into autosomal, sex-linked and antigenic/immunomodulatory specific loci. Functionality of loci was established through translation and BLAST search using the WormBase Parasite database [54]. The antigenic and immunomodulatory specific loci were selected according to their availability in the RAD-loci dataset and extensive literature search consisting of; tegumental antigens [55–57], Tetraspanin proteins [58–60], Venom-like allergen proteins [61,62], Invadolysin [26] and excretory antigenic proteins [60].

### Predicting genomic outliers associated with specific species

Loci included in this analysis were present across *S. bovis*, *S. haematobium* and *S. guineensis* species and present in all individuals within each species population, resulting in a final set of 17,892 loci. A global outlier loci analysis was conducted using the program BAYESCAN v. 2.1 [63], with default conditions using loci with a minor allele frequency (MAF) of 0.05, with all loci that had a probability higher than $p = 0.5$ were treated as putative outliers [64].

Outlier loci that were differentially fixed between species were individually checked against the WormBase Parasite database [54]. This was performed to ascertain whether they were present in transcribed or non-transcribed genomic locations. If they were part of a transcribed gene, the gene name was recorded (S2 Table) and the function was checked on the National Centre for Biotechnology Information (NCBI) gene database (https://www.ncbi.nlm.nih.gov/). The function and cellular location of the gene and protein product was then checked using the uniprot protein database (https://www.uniprot.org/). To further visualise contrasting species and whether genomic regions showed evidence of selection in the global analysis, RAD loci were indexed with positional information and their $F_{st}$ values were plotted along the genetic map.

## Results

### Genetic comparison between species

Genetic clustering analysis using STRUCTURE strongly supported three genetic clusters (Figs 2 and S1 and S2). These genetic clusters were consistent with the three species and admixed population, where levels of introgression varied between the clusters. The highest degree of genetic distinctiveness was exhibited by the *S. bovis* ($Q_{S.\ bovis}$ = 100%) and *S. guineensis* isolates ($Q_{S.\ guineensis}$ = 0.997% and $Q_{S.\ haematobium}$ = 0.003%). The *S. haematobium* isolates displayed a consistent degree of introgression of *S. bovis* ($Q_{S.\ haematobium}$ = 0.983% and $Q_{S.\ bovis}$ = 0.017%). In comparison the genotypes that were previously flagged as being from hybrid populations exhibited the highest degree of admixture within the samples analysed ($Q_{S.\ haematobium}$ = 0.435, $Q_{S.\ guineensis}$ = 0.563, $Q_{S.\ bovis}$ = 0.002) (S1 Fig).

In total, all four individuals from Cameroon were genetically assigned to the admixed population cluster *(Hy1, Hy2, Hy3* and *Hy4)*. Admixed individuals from the Kumba (1849) region of Cameroon were previously identified as *S. haematobium* through egg morphology [34]. Genetic assignment of introgressive *S. bovis* loci within the *S. haematobium* genetic cluster was

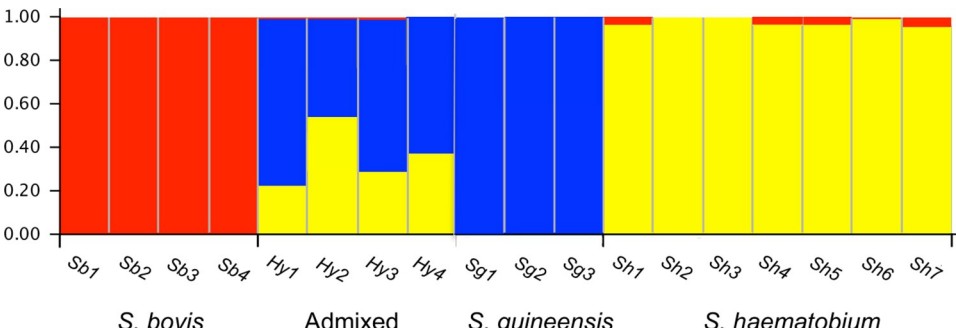

**Fig 2. Output from STRUCTURE analysis using K = 3, see S3 Fig for comparisons with K = 1–10.** Each bar represents an individual and the relative proportion of its genotype belonging to each genetic cluster. The analysis was performed excluding differentially fixed SNPs, outlier loci and adjusted for computational power. Total number of loci used in the analysis was 10,000.

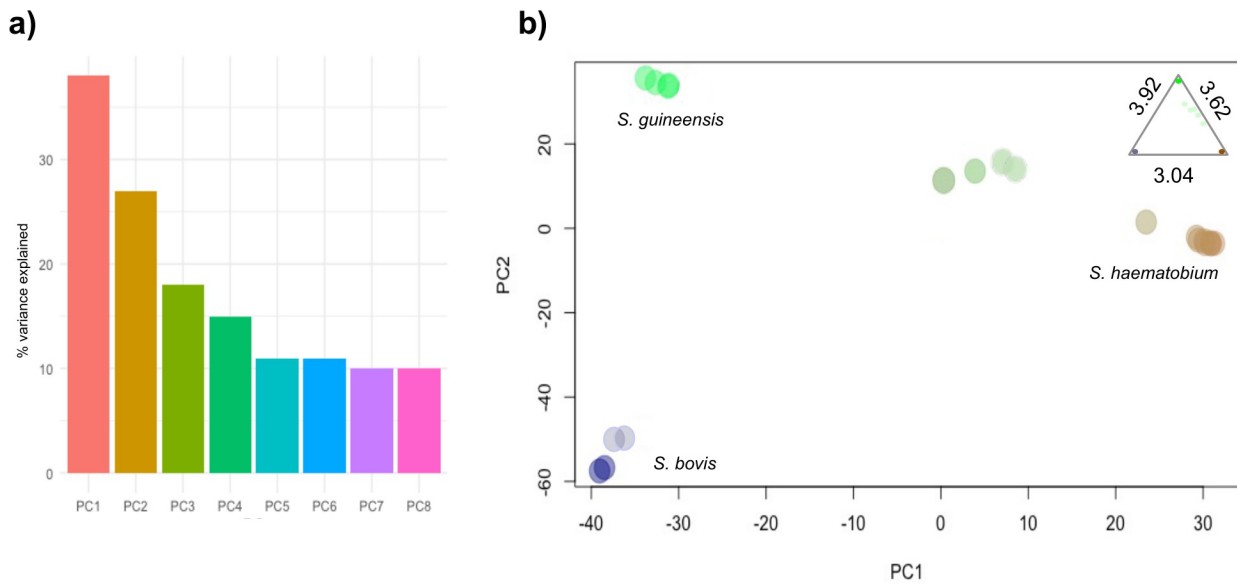

**Fig 3. A)** Multivariate analysis illustrating the genetic differentiation between the isolates. Analysis was performed with 19 isolates showing relative percentages of variance explained by the first eight principal components **B)** PCoA plot of each species with average Euclidean distance of clusters in top right hand corner.

present in 5 (*Sh1*, *Sh4*, *Sh5*, *Sh6* and *Sh7*) of the seven samples. Estimates of heterozygosity corroborated the observed patterns of hybridisation within the admixed isolates; with significantly higher vales compared to other species clusters (Tukey's p < 0.00001, S2 Fig).

The ML analyses of the respective SNP dataset produced consistent phylogenetic grouping of *Schistosoma* species and hybrid isolates respectively (S3 Fig). Branch support was strong throughout topology, indicating a sister-group placement of hybrid individuals alongside *S. haematobium*.

Consistent with the STRUCTURE analysis, the PCoA returned 4 separate clusters, consistent with the three separate species and the admixed hybrid population. The *S. haematobium* isolates were clearly differentiated from S. *guineensis* and S. *bovis* on the axis that explained the majority of the genetic variation (PC1, 38%) (Fig 3A-3B).

## Predicting candidate genes association with speciation

Analysis for potential outlier genes was based on *S. bovis* x *S. haematobium*, *S. haematobium* x *S. guineensis* and *S. bovis* x *S. guineensis* species comparisons revealing a total of 355 loci having a >50% probability of being outliers based on comparison across all three species (Fig 4). Numbers of outlier loci varied per pairwise species comparison; *S. bovis* x *S. haematobium* cross indicating the highest number of outlier loci (153), with *S. haematobium* x *S. guineensis* having the lowest number of outlier loci (93). The greatest proportion of outlier genes was located on the Z sex chromosome (47: *S. haematobium* x *S. bovis*, 25: *S. haematobium* x *S. bovis*, 30: *S. guineensis* x *S. bovis*) (S2 Table).

Out of the complete dataset of outlier genes, a proportion of the genes were successfully translated and annotated for functionality (*S. bovis* x *S. haematobium*: 51, *S. haematobium* x *S. guineensis*: 29, *S. guineensis* x *S. bovis*: 43) (Fig 4A, 4B and 4C). Although there was an overall mixed biological and molecular functionality of all translated outlier loci, when genes were mapped to cellular location (S4 Fig), the largest proportion were placed within the cell membrane (*S. haematobium* x *S. bovis*: 36%, *S. guineensis* x *S. haematobium*: 34% and *S. guineensis* x

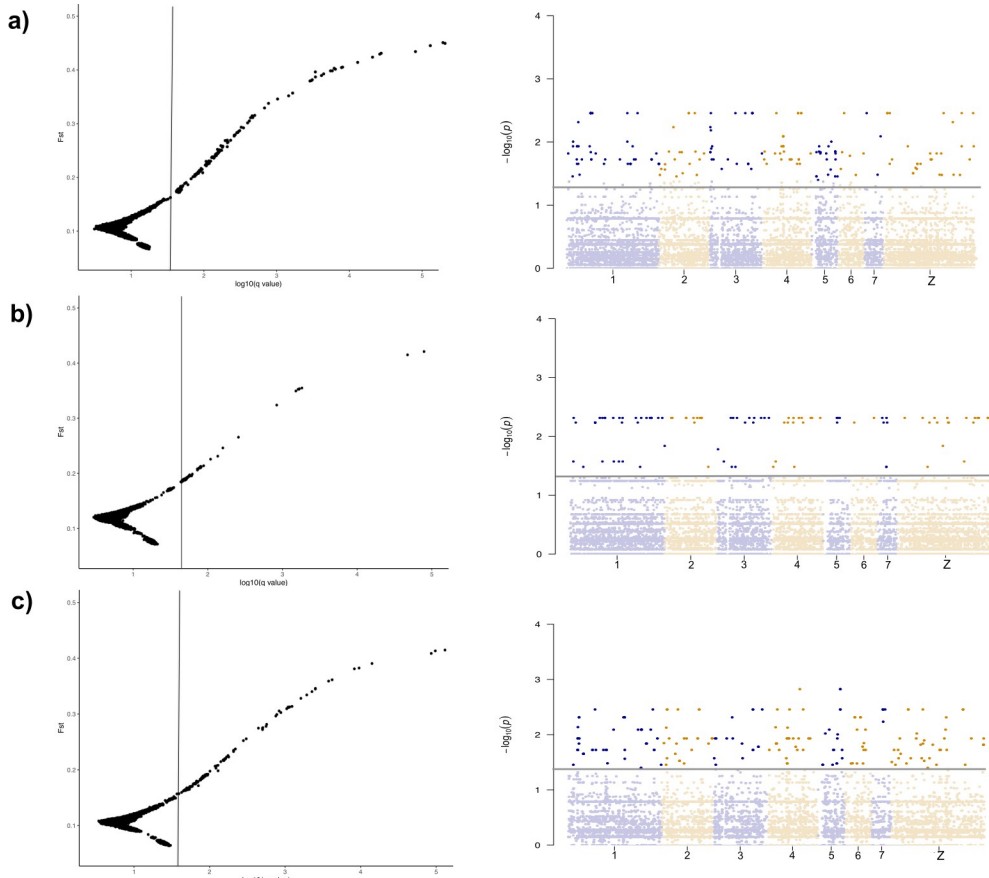

**Fig 4. Plot of $F_{st}$ outliers and genome scan of all identified loci that match to the indexed *S. mansoni* genome.** On the left $F_{st}$ values obtained from BAYESCAN, each plot figure represents outliers in each simulated comparison: **A)** *S. haematobium* x *S. bovis*, **B)** *S. haematobium* x *S. guineensis*, **C)** *S. guineensis* x *S. bovis*. Loci that fall to the right of the vertical line have a probability greater than 50% of being designated outlier loci. Genome scan plots on the right show genomic location of outlier loci across the genome.

*S. bovis*: 32%) (S2 Table). The molecular functionality of outlier loci mapped to the *Schistosoma* genome indicated a consistent presence of loci associated with proteins involved in calcium channel regulation/transportation (*S. haematobium* x *S. bovis*: 7, *S. guineensis* x *S. haematobium*: 6, *S. guineensis* x *S. bovis*: 4) and ATP binding (*S. haematobium* x *S. bovis*: 6, *S. guineensis* x *S. haematobium*: 7, *S. guineensis* x *S. bovis*: 11) (S4 Fig).

## Detecting genes under introgression

In order to identify genes with extreme patterns of introgression, we used the Bayesian genomic cline model of Gompert and Buerkle (2011) implemented in bgc [36]. The comparisons were based on parental *S. guineensis* and *S. haematobium* loci, with assigned admixed isolate from previous cluster analysis. We defined genes as having excess patterns of introgression if their posterior 95% credible intervals for α or β were greater, or less than zero (Fig 5A, 5B and 5C). Positive α values indicate an excess probability of *S. haematobium* ancestry while negative α indicates an excess probability to *S. guineensis* ancestry. The genomic cline data sets were divided into loci located across sex chromosomes, autosomes and antigenic proteins. Of the sex-linked loci within the admixed individuals, 60 loci from a possible 275 (21.8%) were significantly underrepresented with regard to heterozygotes, with their steep cline indicating a rapid

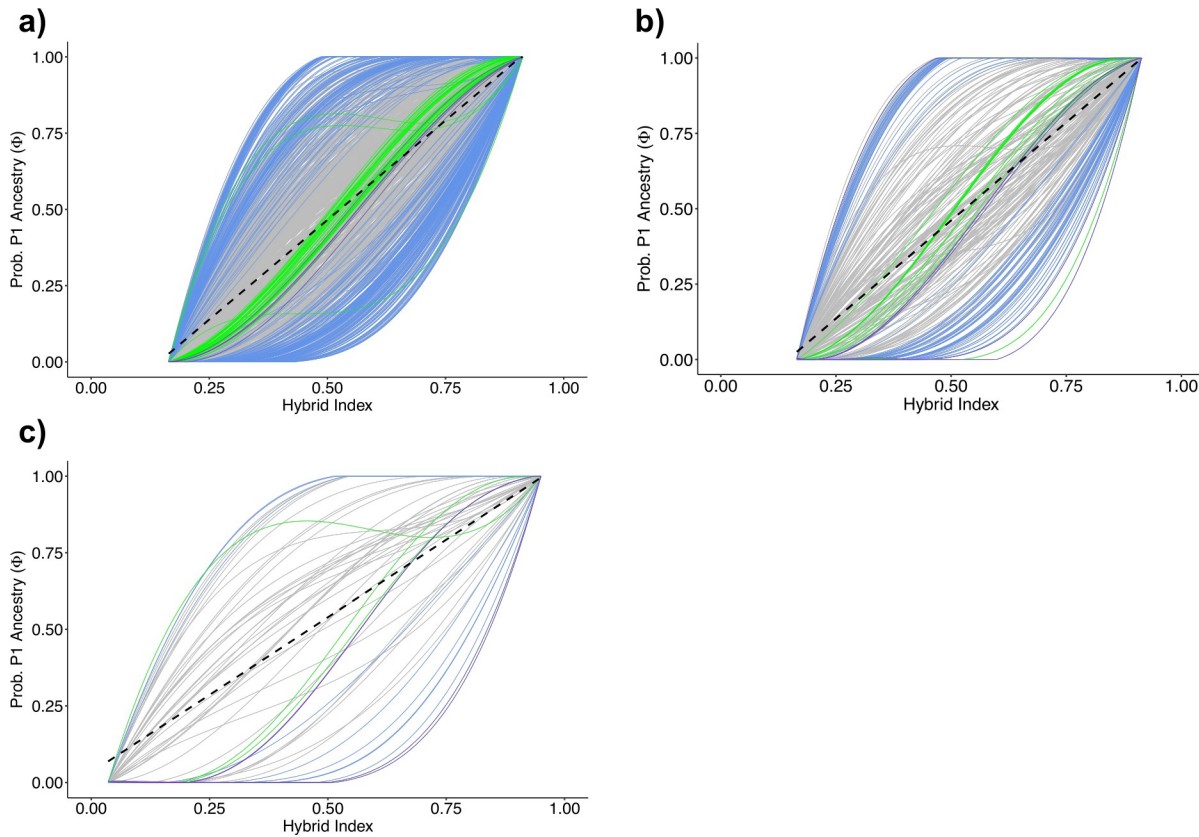

**Fig 5.** Introgression of **A)** Autosomal, **B)** Sex-linked and **C)** Antigen derived candidate gene sets. Genomic clines of candidate genes with excess ancestry, the grey shaded region represent the neutral genome background and are bounded by the positive and negative means of α. The (blue) colour lines represent outlier locus clines with evidence of *S. guineensis* (top left) and *S. haematobium* (bottom right) ancestry. Green lines represent genes that present heterozygous advantage, with shared ancestry between two parental species. When both $\alpha_{ij}$ (genomic cline center) and $\beta_{ij}$ (genomic cline rate) are both set to 0, this describes a on-to-one relationship of the hybrid index, describing neutral ancestry expectations.

transition among parental genotypes (13: *S. guineensis* and 47: *S. haematobium*) (S3 and S4 Tables). There were a total of six β outliers, with three indicating steep negative clines and thus restricted introgression. Autosomal-linked loci included a dataset of 1009, of which 167 loci (16.5%) indicated excess ancestry deviating from neutral expectations (37: *S. guineensis* and 129: *S. haematobium*) (S2 Table).

The antigen loci dataset consisted of 23 antigen-associated loci. These were collated using a literature search to ascertain potential genes that may infer antigenic immunomodulation, 10 loci demonstrated characteristics deviating from neutral expectations, revealing excess parental ancestry (4: *S.* guineensis; TSP-23 [58], Smp_179010, Smp_123540 [61], SMTAL11 [55] and 10: *S. haematobium*; Sm-22.6, Smp_344610 [55], Smp_StoLP [63], Smp_327920 [61], Smp_150960 [26], Smp_344800, Smp_SMTAL13 [56], Smp_308310 [63], Smp_320450, Smp_194970 [59]). Out of the 24 loci, one indicated a cline rate outlier (β) with steep clines (S5 Table).

## Discussion

### Introgression occurrence

The RADseq data presented here confirms the existence of *S. guineensis* x *S. haematobium* hybrids in Cameroon at the time of sample collection. While the data support previous

morphological [29,30,32] and Sanger sequencing methodologies [33,34], a genome-wide dataset reveals greater definition. This has improved our current knowledge regarding the introgression and differentiation that exists across the populations, where rDNA nuclear and mitochondrial markers are limited.

The confirmed presence of *S. guineensis* and *S. haematobium* admixed population supports the hypothesis of long-term introgression occurring between species of the *S. haematobium* group in Cameroon [16,34,35] (Fig 2). The precise timing of the hybridisation event cannot be ascertained however, owing to the fact that the proportional introgression is not directly 50%; the samples are not F1 hybrids. Although the isolates are not direct F1 offspring, comparison with the introgression evident in *S. bovis* x *S. haematobium* hybrids (isolates; *Sh1*, *Sh4*, *Sh5*, *Sh6* and *Sh7*), the admixed population appears to be in an early stage. Introgression of *S. bovis* was also present within the admixed isolate dataset; this could be owing to the consistent presence of *S. bovis* ancestry in potential *S. haematobium* parental species [65] (Fig 2). Our results confirm previous genome-wide hybridisation studies in *Schistosoma*, whereby regional hybridisation is evident in varying degrees across Africa. However, this has only been examined between *S. haematobium* and *S. bovis* [26,66]. Pairwise genomic comparison between the two species revealed large regions of the two genomes displaying high similarity (>99%) spanning up to 100kb [67]. This leads to the assumption that remnants of an ancient hybridisation event between *S. haematobium* and *S. bovis* are widespread in populations of *S. haematobium* across mainland Africa. This is not the same pattern that exists between species of *S. guineensis* and *S. haematobium*. No introgression was revealed to exist in *S. guineensis* from São Tomé and Príncipe, implying potential refugia of the species existing on the islands, with geographic division remaining a primary barrier to introgression. Recent infection mapping on the island revealed no infective presence of urogenital schistosomiasis, and thus indicates this may still be the case [68]. Small areas of *S. guineensis* infection do still exist in select regions in Cameroon at low levels, primarily within the South West of the country [69]. This suggests that hybridisation between the two species remains a geographically specific phenomenon within sympatric populations, rather than introgressive remnants existing within extant *S. guineensis* localities. The rarity of said events would be associated with anthropogenic and environmental risk factors that would increase the likelihood of snail and definitive host intermixing. In respect to Cameroon, local deforestation has been implicated in a factor to allow the intermediate snail host *Bulinus truncatus*, the intermediate host of *S. haematobium* to become established, resulting in sympatric populations with *S. guineensis* [70]. Although the RADseq data has confirmed hybridisation and introgression, it is most likely underrepresented as RADseq operates as a reduced representation of the overall genome. A wider geographical surveillance survey would ascertain the current genomic constitution of hybrids populations in the region, as well as confirming continued existence of *S. guineensis* in São Tomé.

### Identification of taxon-specific outliers

To address one of the major hypotheses in this paper, we were interested in identifying genome-wide loci that exhibited patterns likely driven by selection to determine potential species-specific adaptive loci (Figs 4 and S4 and S2 Table). Across the loci dataset, the heterogeneous genomic landscape varied in allelic distribution, as seen in other genome-wide data sets for *Schistosoma* [26,67]. The distribution of outliers across the genome included loci with extreme allelic differentiation that are linked to genomic regions under selection. This approach can account for realistic ecological scenarios where effective population and sample sizes may differ. This study found that loci displaying extreme divergence traits were relatively

rare in comparison to the entire loci set, suggesting the importance that these outlier loci are under selection.

Natural hybridisation of *Schistosoma* species within the *S. haematobium* group indicates that species barriers that do exist are primarily associated with overlapping of snail host populations, where given the opportunity, hetero-specific crosses will lead to the formation of viable hybrid zones in the environment. One of the major benefits demonstrated here with using RADseq techniques over traditional microsatellite or single nuclear/mitochondrial markers is the ability to highlight genome-wide variation and differentiation between species.

The maintenance of speciation in sympatric foci could result from three mechanisms; 1) temporal or spatial separation, restricting gene -flow in natural population conditions; 2) frequency-dependent encounter rates influencing mate choice of adult worm pairs; and 3) hybrids between ecotypes demonstrating a higher fitness cost as a result of post-zygotic species barriers. The role that physical separation of *Schistosoma* host populations plays in maintaining genetic divergence has been discussed previously, with definitive host movement [10] and irrigation measures leading to intermediate snail species translocation [5] being implicated as a factor in the breakdown of species barriers. However, there is some evidence to suggest parallel speciation can exist between members of the *S. haematobium* group, where no introgression has been detected in *S. haematobium* in Zanzibar [26], despite the presence of *S. bovis* being recently discovered on the island [68]. This suggests that host interactions may in fact be the primary barrier in maintenance of species boundaries. With both *S. guineensis* and *S. haematobium* infecting human definitive hosts, both species would see a greater frequency of interaction compared to the livestock species *S. bovis*. Within laboratory settings, experimental co-infections has indicated *S. haematobium* shows dominance over *S. guineensis* males, influencing mate choice encounter rate, reduced *S. guineensis* fitness [13,27,67,69], reducing the opportunity for full reproductive isolation in *S. guineensis*.

Intrinsic post-zygotic barriers, such as non-viable hybrid populations or maladaptation of hybrids to host environments, may contribute to any genetic divergence which exists in the *S. haematobium* group. Mixed species worm pairings within a definitive host would need to demonstrate the ability to survive host immune response, crucial to avoid incipient speciation occurring in a population. The adaptive outlier loci highlighted within our analysis indicate an overall mixed functionality of genes that demonstrate greater degrees of divergence between species. When examining the cellular location of outlier loci in the dataset, 31 of 123 were associated with membrane-bound proteins (S3 Fig). The biological functionality revealed a consistent presence of loci that are directly or indirectly associated with calcium-gated ion channels or calcium ion transport (S1 Fig) $Ca_v$ channels initiate the contraction of the musculature, intrinsically linked to neuropeptide based signalling, crucial to the neuromuscular system of mature schistosome worms [70]. The effect that the calcium gated ion channels have on the survival of parasitic flatworms has been noted, with numerous anthelminthic drugs having an agonistic affect on gated ion channels, leading to a lethal cellular influx of ions [71,72]. The current drug of choice for treatment of *Schistosoma* infections is praziquantel, where the links have been established between drug resistance and calcium gated channel α1/ β subunits [71]. Loci that are associated with survivability would present greater adaptive importance and play a role in the viability of any hybrid population. This is also of importance when considering the definitive host occupied by each species, whereby outliers show similar biological functionality in human infecting species comparisons (*Sg x Sh*) as ruminant/humans comparisons (*Sb x Sh* and *Sg x Sb*). Loci that are highlighted here infer increased survivability irrespective of definitive host adaptation.

### Functional genomic architecture within the Cameroon hybrid zone

Regardless of the biological scenarios involved in the inter-specific mixing of *S. haematobium* and *S. guineensis*, a clear hybridisation hotspot existed in Cameroon [28–31]. As within the divergence outlier analysis, the genome-wide landscape of introgression within *S. guineensis* and *S. haematobium* varies considerably, containing signals of extreme loci-specific clines that lie outside the expectations of neutral introgression. For these loci, the expectation is that selection has acted against gene flow to maintain or resist allelic inheritance from one parental species more than expected in neutral conditions [34]. Loci that exhibit these steep sigmoidal clines (Fig 5) may be characterised as producing an increase in hybrid population fitness, or inversely are not inherited by the hybrid as it is selected against due to hybrid incompatibility or non-viability [34].

Our evidence presented here provides a link to patterns of genome-wide introgressive dominance with phenotypic traits expected to play a crucial role in hybrid fitness and infection success (S2–S4 Tables). The higher percentage of the *a priori* set of tegumental antigens selected from the data (41.7%) indicates excess ancestry from *S. haematobium* compared to a lower percentage of autosomal and sex-linked loci. The biological importance of antigens would indicate an increased propensity to undergo significant selective pressure driven by the host immune system. The tegument of schistosomes possesses bound transmembrane proteins that directly interact with the host immune system, alongside the release of immunomodulatory proteins that aid in the successful infection of the host. Four loci potentially demonstrating such bias toward one parental line relate to the parasite protein family Tegument-Allergen-Like (TAL) (Sm-TAL11, Smp_335630, SMTAL13, SMTAL5) (Fig 5C and S5 Table). These proteins are all involved in down regulation of IgE and IgG4 activity via the immune system [52,54]. Two more loci were associated with Venom Allergen-Like antigens (VAL proteins) (Smp_123540; SmVAL12, Smp_300070; SmVAL7). The VAL protein family is made up of 29 secretory proteins that bind or disrupt numerous host immune interventions to mitigate infection [58,59].

The relationship between genomic outliers and antigens could demonstrate antigenic adaptation in hybrid progeny, influencing infective success. With both species infecting humans, the excess inheritance of *S. haematobium* antigens would infer an increased significance of immunomodulatory capacity of said proteins in respect to infective success. The general consensus of antigen inheritance within a parasite population would be maintenance of a heterozygote advantage, where a diverse antigenic profile of the parasite would be advantageous in combating host immune response [73]. Dominant inheritance of certain antigens from particular parental species would have implications for the maintenance of *Schistosoma* species boundaries, shifting the adaptive landscape of *Schistosoma* populations. This would potentially promote hybrids that display an increased infective success compared to the incumbent population. This would be particularly evident with an incumbent species that exhibits reduced effective population size; here genetic swamping may take effect, whereby there would be a replacement of local genotypes with a population that displays a greater degree of infective success [74]. This would be of heightened importance when considering potential disease control interventions; especially where anthropogenic and environmental factors affect movement of *Schistosoma* into a region that previously had displayed low prevalence or non-human infecting species [75]. This is of particular importance when considering an epidemiological shift from intestinal to urogenital schistosomiasis and the increase transmission potential with urinary spread.

## Conclusion

The epidemiological dynamics of schistosomiasis and propensity of hybridisation in Cameroon have already been well documented. The data presented here offers genomic evidence as to how the genomic swamping of hybrids occurs within these regions, with the eventual replacement of *S. guineensis* over a period of approximately 25 years. Our study suggests that the tegumental antigens and proteins, which play a key role in the survivability of mature worms, act as crucial adaptive factors, increasing the viability of hybrid populations. A translocation event of *S. haematobium* has been shown here to be invasive across Cameroon, highlighting the importance for regular genetic surveillance of *Schistosoma* populations, particularly in areas where interventions are aimed at elimination. Here is a prime example of parasite spread being governed by definitive host behaviour, and similar irrigation and water management approaches to those in Cameroon could lead to further hybrid populations in other endemic regions elsewhere.

Although samples were limited, our results demonstrate that utilising RADseq population genomic methods can clearly identify population structuring and detect introgression, whereas traditional Sanger sequencing methods are inadequate to do so [4,34,76]. We also show the ability to highlight specific loci prone to directional introgression and selection that may increase the ability of certain species or hybrid of parasite to proliferate over others, within an endemic region.

## Supporting information

**S1 Fig. Likelihood estimates for the number of genetic clusters across all 19 isolates.** Likelihoods were derived from four independent runs per and tested for up to 10 genetic clusters (K = 1–10). Plot shows the log likelihood values as ΛK produced by STRUCTURE. The four genetic clusters result in the highest log likelihood values, lowest standard deviation and highest ΛK.
(TIFF)

**S2 Fig. Estimates of heterozygosity ($H_{obs}$; proportion of heterozygous RAD loci) compared between all species genetic clusters.**
(TIFF)

**S3 Fig. The highest scoring maximum likelihood tree estimated using RAxML using 10,000 unlinked SNP's.** Branch support figures addressed above nodes.
(TIFF)

**S4 Fig. Pie chart describing cellular location of outlier loci as described by BAYESCAN in each simulated cross.** The translation and functionality is established via the WormBase ParaSITE database. Each pie chart describes cellular locations of outliers described in species comparisons across; **a)** S. haematobium x S. bovis, **b)** S. haematobium x S. guineensis, **c)** S. guineensis x S. bovis.
(TIFF)

**S1 Table. Sample table of isolates used within the study including: SCAN sample ID, initial species identification at collection, RADseq assigned species ID, date of collection and samples site.**
(TIFF)

**S2 Table. Data table presenting the outlier analysis of species comparisons: S. bovis x S. haematobium, S. haematobium x S. guineensis, and S. guineensis x S. bovis.** Table presents Loci number, genomic sequence, translated sequence, biological and molecular function of

each loci highlighted in BAYESCAN outlier analysis.
(XLSX)

**S3 Table. Data table presenting the output from Bayesian Genomic Cline analysis for Autosomal Loci alongside alignment and translational loci outcome.**
(XLS)

**S4 Table. Data table presenting the output from Bayesian Genomic Cline analysis for the W chromosome linked loci alongside alignment and translational loci outcome**
(XLSX)

**S5 Table. Data table presenting the output from Bayesian Genomic Cline analysis for the Antigen specific loci alongside alignment and translational loci outcome from**
(XLS)

## Acknowledgments

We would like the thank the Natural History Museum, London for the genetic material provided via the SCAN archive, it was curated and collected in a collaborative effort and we acknowledge the support and generosity of the partnerships involved in the collections, particularly in relation to the endemic countries involved, and past colleagues that maintained the live material.

## Author Contributions

**Conceptualization:** Toby Landeryou, Fiona Allan, Bonnie L. Webster, Aidan M. Emery.

**Data curation:** Toby Landeryou, Muriel Rabone, Rosie Maddren, Louis-Albert Tchuem-Tchuenté.

**Formal analysis:** Toby Landeryou, Rosie Maddren.

**Funding acquisition:** Fiona Allan, David Rollinson, Aidan M. Emery.

**Investigation:** Toby Landeryou, Muriel Rabone, Fiona Allan, Aidan M. Emery.

**Methodology:** Louis-Albert Tchuem-Tchuenté.

**Project administration:** Louis-Albert Tchuem-Tchuenté, Roy M. Anderson.

**Resources:** Aidan M. Emery.

**Software:** Toby Landeryou.

**Supervision:** Aidan M. Emery.

**Validation:** Toby Landeryou, Rosie Maddren, Roy M. Anderson.

**Visualization:** Toby Landeryou, Muriel Rabone.

**Writing – original draft:** Toby Landeryou.

**Writing – review & editing:** Toby Landeryou, Muriel Rabone, Fiona Allan, Bonnie L. Webster, Aidan M. Emery.

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
