## [Decision Letter · Decision Letter 0]

23 Sep 2021

Dear Dr Landeryou,

Thank you very much for submitting your manuscript "Genome-wide insights into adaptive hybridisation across the Schistosoma haematobium group in West and Central Africa" for consideration at PLOS Neglected Tropical Diseases. As with all papers reviewed by the journal, your manuscript was reviewed by members of the editorial board and by several independent reviewers. The reviewers appreciated the attention to an important topic. Based on the reviews, we are likely to accept this manuscript for publication, providing that you modify the manuscript according to the review recommendations. 

Sincerely,

Keke C Fairfax, PhD

Deputy Editor

Keke Fairfax

Deputy Editor

Reviewer's Responses to Questions

**Key Review Criteria Required for Acceptance?**

**Methods**

-Are the objectives of the study clearly articulated with a clear testable hypothesis stated?

-Is the study design appropriate to address the stated objectives?

-Is the population clearly described and appropriate for the hypothesis being tested?

-Is the sample size sufficient to ensure adequate power to address the hypothesis being tested?

-Were correct statistical analysis used to support conclusions?

-Are there concerns about ethical or regulatory requirements being met?

Reviewer #1: Please see section general comments

Reviewer #2: -Are the objectives of the study clearly articulated with a clear testable hypothesis stated? Yes

-Is the study design appropriate to address the stated objectives? Yes

-Is the population clearly described and appropriate for the hypothesis being tested? Yes

-Is the sample size sufficient to ensure adequate power to address the hypothesis being tested? Yes

-Were correct statistical analysis used to support conclusions? Yes

-Are there concerns about ethical or regulatory requirements being met? No ethical issue found

Since this study used archived samples that were collected in different time periods, I'm wondering how this issue affects the analysis pipeline, results, and interpretation.

**Results**

-Does the analysis presented match the analysis plan?

-Are the results clearly and completely presented?

-Are the figures (Tables, Images) of sufficient quality for clarity?

Reviewer #1: Please see section general comments

Reviewer #2: -Does the analysis presented match the analysis plan? Yes

-Are the results clearly and completely presented? Yes

-Are the figures (Tables, Images) of sufficient quality for clarity? Yes

Fig 5 looks fancy, however, it's difficult to interpret for a general audience. Therefore, I would suggest it might be better to move figure 5 to supplementary data and move the S 4 Table (showing list of identified antigens/proteins), which is more obvious and informative to the main table of the manuscript.

**Conclusions**

-Are the conclusions supported by the data presented?

-Are the limitations of analysis clearly described?

-Do the authors discuss how these data can be helpful to advance our understanding of the topic under study?

-Is public health relevance addressed?

Reviewer #1: Please see section general comments

Reviewer #2: -Are the conclusions supported by the data presented? Yes

-Are the limitations of analysis clearly described? Yes

-Do the authors discuss how these data can be helpful to advance our understanding of the topic under study? Yes

-Is public health relevance addressed? Yes

**Editorial and Data Presentation Modifications?**

Reviewer #1: NA

Reviewer #2: Line 186-187 Please clarify this statement "DNA was sequenced from a total of 18 worm pairs were separated, with DNA

extracted from individual males"

Line 323 This is no Fig3c

Line 641-2 unpolished writing

**Summary and General Comments**

Reviewer #1: Congratulations to this interesting article on possible hybridisation events across the S. haematobium group in Western and Central Africa. The manuscript reads well throughout though some sections starting with the introduction would benefit from some English editing as they are not easy to understand, which impacts the subsequent flow of the manuscript. For instance, L85-86, I believe S. haematobium is reported to cause urogenital schistosomiasis disregarding of whether it occurs in an endemic or non-endemic area. Consider explaining to readers new in this field what hybridisation or introgression actually means also from a genomic point of view. Also, the potential impact of schistosomal hybridisation on control and prevention measures isn't explained sufficiently to highlight the importance/concerns of such hybrids. Are they not treatable with PZQ? 

Most importantly as this is the backbone of all your analyses, please explain in more detail your source isolates, i.e. field isolates from individuals and snails from which parts of the countries listed, single species isolates, admixed isolates. To which schistosomal stage does the term "isolate" refer to? Why did you look into adult males only? Also, some literature states that larval stages passaged through laboratory intermediate and definitive hosts is prone to artificial selection of specific genotypes. Please put more emphasis on describing your source isolates and if possible simplify the description of your analytical part.

Reviewer #2: This is the important study that will provide the new knowledge to the field. It took one more step ahead by looking at a genome-wide structure of the hybrid species to visualize the underlying mechanism of hybridization. In addition, the authors also provide the functional genomics structure of the hybrids, which allow us to understand the benefit of hybridization in worm survival. Overall, I really enjoyed reading this manuscript.

PLOS authors have the option to publish the peer review history of their article (what does this mean?). If published, this will include your full peer review and any attached files.

Reviewer #1: No

Reviewer #2: No

Figure Files:

Data Requirements:

Reproducibility:

References

---

## [Editor Report · Decision Letter 1]

11 Dec 2021

Dear Dr Landeryou,

We are pleased to inform you that your manuscript 'Genome-wide insights into adaptive hybridisation across the Schistosoma haematobium group in West and Central Africa' has been provisionally accepted for publication in PLOS Neglected Tropical Diseases.

Best regards,

Keke C Fairfax, PhD

Deputy Editor

Keke Fairfax

Deputy Editor

---

## [Editor Report · Acceptance letter]

17 Jan 2022

Dear Dr Landeryou,

We are delighted to inform you that your manuscript, "Genome-wide insights into adaptive hybridisation across the Schistosoma haematobium group in West and Central Africa," has been formally accepted for publication in PLOS Neglected Tropical Diseases.

Best regards,

Shaden Kamhawi

co-Editor-in-Chief

Paul Brindley

co-Editor-in-Chief
